# Uni-DlLoRA: Style Fine-Tuning for Fashion Image Translation

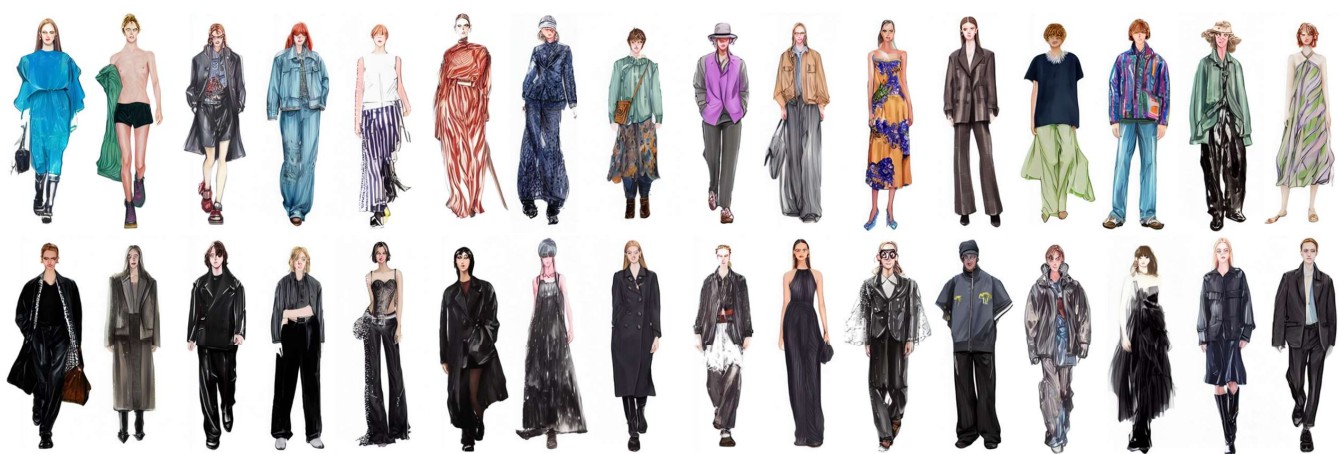

**Figure 1: High-quality fashion illustration generations via the proposed Uni-DlLoRA.**

## ABSTRACT

Image-to-image (i2i) translation has achieved notable success, yet remains challenging in scenarios like real-to-illustrative style transfer of fashion. Existing methods focus on enhancing the generative model with diversity while lacking ID-preserved domain translation. This paper introduces a novel model named Uni-DlLoRA to release this constraint. The proposed model combines the original images within a pretrained diffusion-based model using the proposed Uni-adapter extractors, while adopting the proposed Dual-LoRA module to provide distinct style guidance. This approach optimizes generative capabilities and reduces the number of additional parameters required. In addition, a new multimodal dataset featuring higher-quality images with captions built upon an existing real-to-illustration dataset is proposed. Experimentation validates the effectiveness of our proposed method.

## CCS CONCEPTS

• **Computing methodologies** → **Computer vision tasks**.

## KEYWORDS

fashion synthesis, image-to-image translation, denoising diffusion probabilistic models

**ACM Reference Format:**
Anonymous Author(s). 2018. Uni-DlLoRA: Style Fine-Tuning for Fashion Image Translation. In *Proceedings of Make sure to enter the correct conference title from your rights confirmation emai (Conference acronym 'XX).* ACM, New York, NY, USA, 10 pages. https://doi.org/XXXXXXX.XXXXXXX

**Unpublished working draft. Not for distribution.**

## 1 INTRODUCTION

The advancement of generative models has revolutionized the field of computer vision, particularly in fashion, where the creation and manipulation of images play a pivotal role[2, 10, 18, 42]. Fashion synthesis [8, 19, 20, 53, 56] has emerged as a dynamic area of research, encompassing a spectrum of applications from virtual try-on to appearance and pose transfer. Despite these advancements, the translation of fashion images between distinct domains, such as illustration and realism, remains a challenging topic. This translation is critical for fashion creation and understanding the nuanced interplay between style and content in fashion imagery.

Recent works have begun to explore the synthesis of fashion images, with StylishGAN [58] introducing a dataset that bridges the gap between real and illustrated fashion domains. However, existing methods of fashion image synthesis, while making significant strides, exhibit several limitations: (1) Lack of Dataset Quality: Current fashion illustration datasets often suffer from low resolutions and the presence of backgrounds in real domain images, which hinder the training of models to focus on fashion items exclusively. (2) Inadequate Style Capture: Existing generative models, including diffusion models like SGDiff [46], struggle to accurately capture and replicate the specific stylistic elements of fashion items, particularly when translating between domains with distinct visual characteristics. (3) Limited Style Control: Methods that rely solely on text prompts for style transfer lack precise control over the stylistic nuances, as textual descriptions are insufficient to convey complex visual styles, leading to inconsistent and less realistic outputs. (4) Style Adaptation Challenges: While methods like Lora-Rank Adaptation (LoRA) [13] prevent catastrophic forgetting by using low-rank matrices, they face difficulties in learning specific styles, as the alignment between the condition information and the internal knowledge of the model is not well-established.

To this end, we present Uni-DlLoRA, a novel approach to fashion image-to-image translation that focuses on the fine-tuning of diffusion models for image synthesis and improving style disentanglement. Recognizing the lack of high-quality fashion illustration datasets, we utilized SwinIR [24] and LDSR [36]to perform super-resolution on the StylishU dataset, resulting in a dataset with improved resolution and clarity. The caption of each image is extracted by BLIP [23] and refined by fashion experts for text-conditioning. Our method, Uni-DlLoRA, is designed to address the limitations of current techniques by incorporating image-conditioned information using the proposed Uni-adapter and adapting the UNet denoiser with the Dual-LoRA module to better capture spatial and textural details from both real and illustrative domains. By doing so, Uni-DlLoRA enables the seamless translation of fashion images while preserving their essential visual features and stylistic elements. In addition, we introduce a method for disentangling style from target images or domains and integrating it into source images to achieve stylistic consistency and variety in the generated images. Qualitative and quantitative comparisons with state-of-the-art methods demonstrate the effectiveness of Uni-DlLoRA. All in all, our contribution can be summarized as:

- This article highlights a novel method that fully applies a Uni-adapter to extract latent features from input images and enhances learning in fashion image translation through the novel Dual-LoRA module.
- The article presents a new dataset in response to the existing challenges in the fashion field, which features graphics with better resolution and accurate textual information.
- Additionally, an innovative training method successfully generates images full of detail while effectively disentangling the content and style of the images. Detailed experiments describe the effectiveness and practicality of the method.

## 2 RELATED WORK

### 2.1 Fashion Image Synthesis

Fashion synthesis is a burgeoning research domain within the expansive realm of computer vision. With its formidable generative capabilities, the synthesis models can effectively generate high-quality images based on conditional information. In particular, numerous approaches [4, 9, 22, 55] focus extensively on virtual try-on, a process that involves transferring desired clothing onto a specific person. Other studies [1, 28, 35] concentrate on appearance and pose-guided transfer, wherein the model is capable of transforming the target person to the desired pose based on the given appearance. Recently, image editing has gained popularity, with several methods [16, 17, 52, 56] focusing on the editing of specific elements onto clothing. Some of these methods like SGDiff[46] have achieved significant results through the use of diffusion models, enabling text editing to become a reality. Nevertheless, the translation of fashion images between illustration and real domains remains relatively unexplored compared to other areas within the fashion industry, despite being an important process in fashion creation. StylishGAN [58] first introduced this task into the field of computer vision and developed a dataset containing fashion images from both real and illustrated domains. However, there is still room to improve the quality of the dataset and the generative model.

## 2.2 Fashion Image-to-Image Translation

Image-to-image (i2i) translation is a widely studied and popular research topic, introduced by Isola [15]. The main goal of this task is to accurately and effectively translate an input image into an output image while preserving important visual features and details. This can be used for various applications such as style transfer [7] and image synthesis [54]. Several methods [3, 5, 27] apply a content image and a style reference image to create an image that captures the style of the reference while retaining the content of the original during the generation process. However, the texture and color of the style images are hard to disentangle. Though AAST [14] proposed a model that transfers the images to the target domain while considering the texture and aesthetic, blurred background exists during generation. Other methods [25, 31] tried to transfer the style images to the certain style with pre-trained networks, but they failed to transfer uninformative images [58] to another domain.

Afterwards, text-driven image-to-image translation has gained traction, with several methods [32, 47] achieving significant results by leveraging powerful generation models such as the diffusion model. However, the utilization of text-driven information is constrained in effectively conveying styles or emotions, as objects are easily described, while styles are challenging to articulate in words.

### 2.3 Fine-tuning based on Diffusion Models

The diffusion models [36, 39] have recently gained significant popularity and fine-tuning models based on them are widely used for downstream tasks. However, the over-fitting and mode collapse exists while training the neural network with additional training data. Extensive research paid attention to avoiding such issues. For instance, Dreambooth [38] and Textual Inversion [6] customize the content in the generated image by fine-tuning the image diffusion model with a small set of user-provided example images. However, this approach has a high computational cost, as the entire generating model must be fine-tuned. Lora-Rank Adaptation (LoRA) [13] noted that over-parameterized models exist within a low intrinsic dimension subspace, and thus this method prevents catastrophic forgetting by obtaining information on the parameter offset using low-rank matrices. However, learning specific styles applying LoRA can be challenging. Based on substantial results obtained by adapter methods adopted in pretrained model [34, 44] in several downstream tasks, T2I-Adapter [30] and Controlnet [50] adapt Stable Diffusion to different external conditions and learn the alignment between condition information and internal knowledge, achieving solid results. However, T2I-Adapter finds it challenging to learn the style, while ControlNet struggles to strike a balance between model capability and computational cost.

## 3 METHODOLOGY

### 3.1 Preliminaries

The Stable Diffusion (SD) is a text-to-image model known for its strong performance in generating images from text and images. It comes with pretrained checkpoints, making it the chosen backbone model. The diffusion model consists of two major modules: Autoencoders [48] and a modified UNet [37] denoiser. In the training

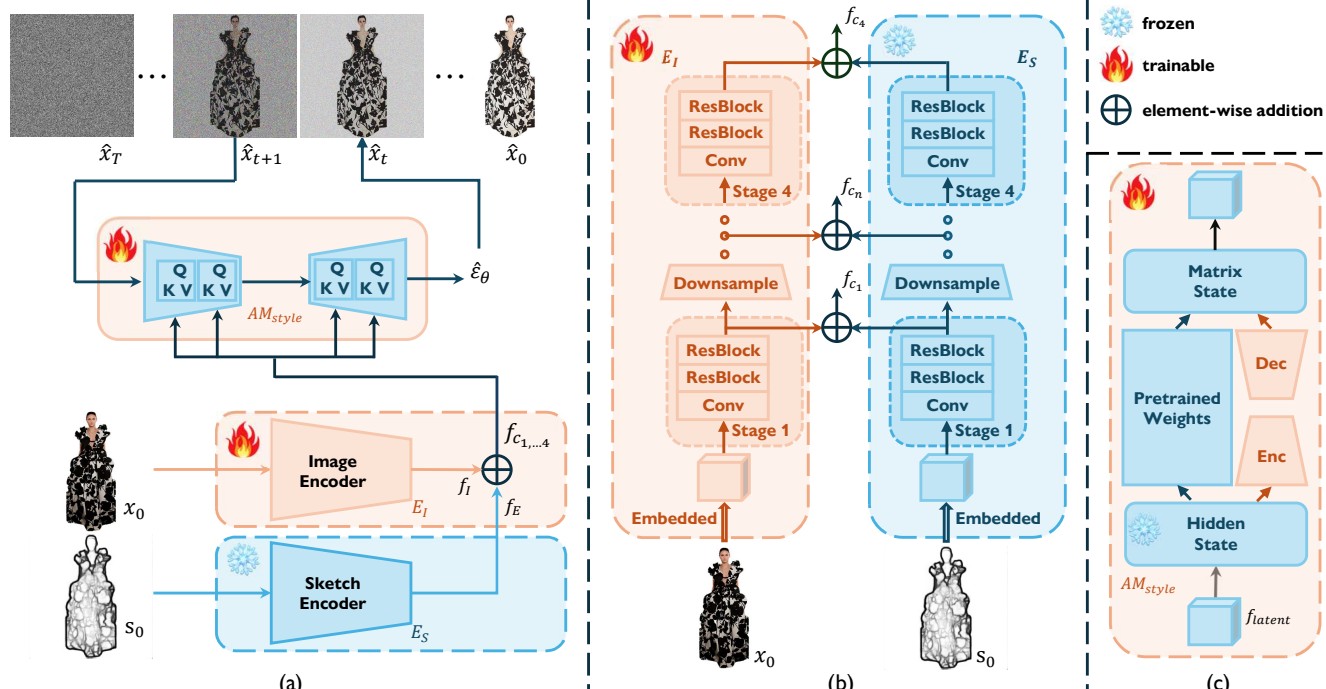

**Figure 2: The proposed Uni-DlLoRA network includes (a) an overview, (b) a detailed process for obtaining mixed conditional embedding from multi-layer features of the image and its sketch, and (c) the specifics of the style adaptation module.**

process, the autoencoder within the whole network will be utilized to encode the images into a latent space, and the latent features will be deliberately noised in a step-by-step manner. After this stage, the modified UNet denoiser is trained to denoise the latent features step by step. The optimization of denoising could be written as:

$$\mathcal{L} = \mathbb{E}_{\mathbf{x}_0, \mathbf{c}, \boldsymbol{\epsilon}, t} \left( \|\boldsymbol{\epsilon} - \hat{\boldsymbol{\epsilon}}_\theta \left( a_t \mathbf{x}_0 + \sigma_t \boldsymbol{\epsilon}, \mathbf{c} \right)\|_2^2 \right), \tag{1}$$

where $\mathbf{x}_0$ denotes the input latent features and $\mathbf{c}$ illustrates the optional conditional informaition. $\boldsymbol{\epsilon} \in \mathcal{N}(0, \mathbf{I})$ represents the added noise and $\mathbf{x}_t = a_t \mathbf{x}_0 + \sigma_t \boldsymbol{\epsilon}$ denotes noised input latent features in step $t$. $\hat{\boldsymbol{\epsilon}}_\theta$ represents the predicted noise from UNet denoiser with conditional informaiton $\mathbf{c}$ according to the Classifier-Free Guidance[12] in the training stage:

$$\hat{\boldsymbol{\epsilon}}_\theta \left( \mathbf{x}_t, \mathbf{c} \right) = \omega \boldsymbol{\epsilon}_\theta \left( \mathbf{x}_t, \mathbf{c} \right) + (1 - \omega) \boldsymbol{\epsilon}_\theta \left( \mathbf{x}_t \right), \tag{2}$$

where $\omega$ is a guidance weight. After the denoising stage, the final image is generated from the cleaned latent features $\hat{\mathbf{x}}_0$ during the decoder part of the Autoencoders. For inference, the latent features $\mathbf{x}_T$, whether originating from random noise or noised input latent features, become progressively clearer as the predicted noise $\hat{\boldsymbol{\epsilon}}_\theta$ is applied at each step $t$ to denoise the latent features, transforming $\mathbf{x}_T$ into $\hat{\mathbf{x}}_0$ with equation:

$$\hat{\mathbf{x}}_{T-1} = \frac{1}{\sqrt{\alpha_t}} \left( \mathbf{x}_T - \frac{1 - \alpha_t}{\sqrt{1 - \bar{\alpha}_t}} \hat{\boldsymbol{\epsilon}}_\theta \left( \mathbf{x}_T, c \right) \right) + \sigma_t \mathbf{z} \tag{3}$$

where $\mathbf{z} \sim \mathcal{N}(0, \mathbf{I})$ denotes the gaussian noise.

To capture the textual information during the denoising stage, the pretrained CLIP [33] is applied to embed text prompts into a sequence of vectors $\mathbf{c}_v$ in the latent space. These vectors are then utilized by the cross-attention module inside the UNet denoiser to

aid in the denoising process. The equation can be written as:

$$CrossAttention \left( \mathbf{q}, \mathbf{k}, \mathbf{v} \right) = \text{softmax} \left( \frac{\mathbf{q} \mathbf{k}^T}{\sqrt{d_k}} \right) \cdot \mathbf{v} \tag{4}$$

where $\mathbf{q} = \mathbf{w}_q \phi(\hat{\mathbf{x}}_t), \mathbf{k} = \mathbf{w}_k \tau(\mathbf{c}_v), \mathbf{v} = \mathbf{w}_v \tau(\mathbf{c}_v)$. $\phi(\cdot)$ and $\tau(\cdot)$ denotes the embedding matrices inside the module and $\mathbf{w}_q, \mathbf{w}_k, \mathbf{w}_v$ represents the weight of projection matrices.

## 3.2 Diffusion Model with Image Conditioned

For the basic diffusion model in the T2I task, the textual information will be embedded firstly into a sequence of vectors in the latent space by pretrained CLIP[33] model and then fed into the cross-attention module inside the UNet denoiser. The generated results are unstable when the input consists solely of text, as text struggles to convey spatial information effectively. The lack of alignment in the results arises from the inherent difficulty of text in offering precise external control. To effectively capture both texture and spatial information, hidden details are extracted from the source image using image and sketch extraction modules, as depicted in Figure 2. Specifically, a novel multi-layer module named Uni-adapter is applied to obtain spatial and texture information, respectively. Inspired by T2I-Adapter [30], the pixel unshuffle [41] operation inside the extraction module is firstly applied to downsample the input. The multi-convolutional layers including two residual blocks are then applied to extract the unshuffled features and multi-scale features will be obtained as: $f_c = \{f_{c_1}, f_{c_2}, f_{c_3}, f_{c_4}\}$. Due to the alignment of latent features from two same-structure extract modules, the equation for the mixed conditional embedding can be written as

**Figure 3: Detailed training process: The mixed conditional embedding is sent to the modified U-Net denoiser for various tasks. An illustration adaptation module is inserted for the synthesis of illustrative images, while a real adaptation module is employed to synthesize real images.**

as:

$$f_{c_i} = \phi_{E_I^i}(\mathbf{x_0}, \theta) + \phi^*_{E_S^i}(\mathbf{s_0}, \theta), i \in \{1, 2, 3, 4\} \quad (5)$$

where $\phi_{E_I^i}(\cdot, \theta)$ and $\phi^*_{E_S^i}(\cdot, \theta)$ represents the multi-scale image and sketch feature extractor, respectively. It is noteworthy that the sketch feature extractor $\phi^*_{E_S^i}$ is fixed with pretrained model while image feature extractor $\phi_{E_I^i}$ is learnable. $\mathbf{x_0}$ denotes the input images and $\mathbf{s_0}$ represents the efficient sketches extracted from input images by the fixed neural network [45], respectively. Inspired by ControlNet [50], zero convolutional layers are adopted in the image extraction module.

## 3.3 Style and Content Disentanglement

The extraction of style from target images or domains, followed by its integration into source images, is significant within the context of the style transfer task. Inspired by Kotovenko et al. [21], two separate style adaption modules named Dual-LoRA were inserted in the UNet denoiser to capture the styles in different domains. As shown in Figure 3 (c), within the Dual-LoRA module, full-rank dense layers that perform matrix multiplication are integrated into the pretrained UNet denoiser to refine the style of the synthesized image. Specifically, the inclusion of parameters in both the image feature extractor and the fixed sketch feature extractor enhances the model's ability to extract spatial and textural information from the input. Specialized style adaptation modules with learnable parameters are inserted into the UNet denoiser to aid in refining the style of the synthesized images, as well as in content and style disentanglement. Unlike simple LoRA [13], two separate style adaption modules within Dual-LoRA are applied to assist specific noise prediction with an equation at each step $t$:

$$\hat{\mathbf{x}}_{r_{t-1}} = \frac{1}{\sqrt{\alpha_t}} \left( \hat{\mathbf{x}}_{r_t} - \frac{1 - \alpha_t}{\sqrt{1 - \bar{\alpha}_t}} \hat{\boldsymbol{\epsilon}}_{\theta r} \left( \hat{\mathbf{x}}_{r_t}, f_c, \theta_r \right) \right) + \sigma_t \mathbf{z}$$

$$\hat{\mathbf{x}}_{i_{t-1}} = \frac{1}{\sqrt{\alpha_t}} \left( \hat{\mathbf{x}}_{i_t} - \frac{1 - \alpha_t}{\sqrt{1 - \bar{\alpha}_t}} \hat{\boldsymbol{\epsilon}}_{\theta i} \left( \hat{\mathbf{x}}_{i_t}, f_c, \theta_i \right) \right) + \sigma_t \mathbf{z} \quad (6)$$

Specifically, the predicted noise in the process can be written as:

$$\hat{\boldsymbol{\epsilon}}_{\theta r} \left( \hat{\mathbf{x}}_{r_t}, f_c, \theta_r \right) = \omega \boldsymbol{\epsilon}_\theta \left( \hat{\mathbf{x}}_{r_t}, f_c, \theta_r \right) + (1 - \omega) \boldsymbol{\epsilon}_\theta \left( \hat{\mathbf{x}}_{r_t}, \theta_r \right)$$

$$\hat{\boldsymbol{\epsilon}}_{\theta i} \left( \hat{\mathbf{x}}_{i_t}, f_c, \theta_i \right) = \omega \boldsymbol{\epsilon}_\theta \left( \hat{\mathbf{x}}_{i_t}, f_c, \theta_i \right) + (1 - \omega) \boldsymbol{\epsilon}_\theta \left( \hat{\mathbf{x}}_{i_t}, \theta_i \right) \quad (7)$$

where $\hat{\boldsymbol{\epsilon}}_{\theta r}$ and $\hat{\boldsymbol{\epsilon}}_{\theta i}$ denotes the predicted noise for real style and illustration style images reconstruction, respectively. $\boldsymbol{\epsilon}_\theta(\cdot, \theta_r)$ and $\boldsymbol{\epsilon}_\theta(\cdot, \theta_i)$ represent the basic UNet denoiser adding real-style adaption module and illustration-style adaption module, respectively.

## 3.4 Training Objectives

As discussed in Section 3.1, the diffusion algorithm progressively adds the Gaussian noise into the original image $\mathbf{x_0}$ with $t$ times and obtains noisy image $\mathbf{x}_t$. The diffusion models will implicitly learn to reconstruct an image from the noisy image by predicting the added noise depending on the timestep $t$ and task-specific conditions $c_t$. During the training process of our proposed method, images in the real domain are utilized as conditions to provide spatial and texture information, as depicted in the figure 3. Given that two separate style adaptation modules are implemented within the UNet denoiser to aid individual noise prediction, a dual loss can be formulated throughout the entire training process as follows:

$$\mathcal{L}^{dual} = \mathbb{E}_{\mathbf{x}_{i0}, f_c, \boldsymbol{\epsilon}_i, t} \left( \|\boldsymbol{\epsilon}_i - \hat{\boldsymbol{\epsilon}}_{\theta i} \left( \mathbf{x}_{it}, f_c, \theta_i \right)\|_2^2 \right)$$

$$+ \mathbb{E}_{\mathbf{x}_{r0}, f_c, \boldsymbol{\epsilon}_r, t} \left( \|\boldsymbol{\epsilon}_r - \hat{\boldsymbol{\epsilon}}_{\theta r} \left( \mathbf{x}_{rt}, f_c, \theta_r \right)\|_2^2 \right) \quad (8)$$

where $\mathcal{L}^{dual}$ is the overall training objective of the entire diffusion model. This objective is directly applied in finetuning diffusion models with an image extractor and Dual-LoRA modules. $\boldsymbol{\epsilon}_i$ and $\boldsymbol{\epsilon}_r$ represent the added noise for images in the illustration domain and real domain, respectively. The parameters within the pretrained UNet denoiser are fixed during the training process.

## 4 EXPERIMENTS

### 4.1 Implementation

**Network Architecture.** Diffusion models denoise the image by applying the conditions from the prompt and the given image. However, the generated image often lacks a strong correlation with the conditional source image owing to the prompt typically not conveying precise semantic information and struggles to perfectly match the spatial and textural details from the image (as shown in Figure 4). Two adapters, namely the image feature extractor and sketch feature extractor, are applied to carry the multi-scale spatial and texture information from size $64 \times 64$ to $8 \times 8$ that match the spatial size of the feature maps inside the UNet denoiser to address this issue. In pursuit of style disentanglement, two distinct style adaptation modules are employed to refine the style of image generation. DDIM [43] is applied to accelerate the process.

**Dataset.** In this study, there are rarely fashion illustration paired datasets. Zou and Wong [58] gathered a dataset StylishU that comprises 3567 paired images consisting of real photos and hand-sketch illustrations. However, the resolution of the images is relatively low, and they contain backgrounds within the real domain images. We initially utilize SwinIR [24] in conjunction with LDSR [36] to perform a super-resolution version StylishU-SR, thereby obtaining images with a resolution of $512 \times 512$. During the training process,

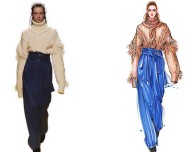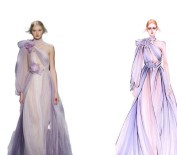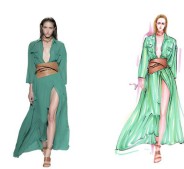

a woman in a blue pants and a **brown** top with a ruffled neckline and a brown top with a blue belt.

a woman in a dress with a flower on the **shoulder** and a long skirt with a bow.

a woman in a **green** dress with a **brown** belt and sandals on her feet, with a white background.

**Figure 4: The dataset includes runway images, paired illustrative images, and captions.**

3467 high-resolution paired images are used as the training dataset, while the remaining 100 paired images are designated as the test dataset. The textual caption of each image is extracted by BLIP [23] and refined by fashion experts for further research.

**Training Details.** The stable-diffusion v1-5 was utilized as the backbone diffusion model. Considering the potential semantic disparity between textual and image information, as shown in Figure 4, **None Prompt** is provided to the UNet denoiser, while the extracted mixed conditional embedding $f_c$ in Equation 5 serves as the sole condition during the training process. The proposed model was fine-tuned on the paired dataset using the AdamW optimizer with a learning rate of $5e^{-6}$. The batch size was set to 8, and the A100 was utilized to train the proposed model for 100,000 iterations. The pretrained PIDNet [49] was employed to extract the sketch from the input images, with the threshold set to 0.5. The parameters of the sketch feature extractor were kept fixed with pretrained weights obtained from training data of COCO17 [26]. Regarding the style adaption modules, the linear encoder-decoder layers with rank=16 are set within the UNet denoiser. To ensure clean background generation, the initial noise will be combined with latent features [29] extracted from images by pretrained Autoencoders.

**Baselines.** The original image is utilized as the conditional information for performing fashion image style transfer. The proposed method is compared with several state-of-the-art methods, including some GAN-based [54] and diffusion-based fine-tuning [13, 30, 50] methods, both qualitatively and quantitatively. The performance of fine-tuned original Stable diffusion (SD)[36] is also evaluated. The test set of the StylishU-SR is applied to the performance of the generated results from each method.

**Metrics.** Following the general practice, four metrics including FID [11], LPIPS [51], CLIP-image [33], and CLIP-aesthetic [40] are applied to evaluate the quality of the generated images for comparison our method with the SOTAs. While the FID score and LPIPS score focus on the latent feature distance between ground truth and generated images, the FID score emphasizes the overall distribution, while LPIPS calculates the distance between each pair of generated images and corresponding ground truth. It is worth noting that, due to the limited number of test datasets, the FID score reported in this article is derived from latent features extracted by the first block of the pretrained CNN, which is denoted as $FID_{64}$. Since this score is based on low-level features, it is more concerned with the similarity between the generated image and the ground truth's underlying features. For these two criteria, the lower the FID and LPIPS scores, the higher the synthesized image quality. Conversely, the CLIP image assesses the cosine similarity between the ground

**Table 1: Qunatitative evaluation and comparison between several SOTA methods with Ours.**

| Methods | Metrics | | | |
|---|---|---|---|---|
| | $FID_{64}$ ↓ | $LPIPS$ ↓ | CLIP-image ↑ | CLIP-aes ↑ |
| CycleGAN | **0.454** | **0.206** | 86.776* | 5.322 |
| SD(add text) | 2.677 | 0.298 | 74.748 | 5.598* |
| LoRA(add text) | 0.605 | 0.233 | 81.530 | **5.638** |
| SD-finetuned | 0.586 | 0.586 | 83.122 | 5.448 |
| ControlNet | 2.078 | 0.216 | 85.863 | 5.415 |
| T2I-Adapter | 0.762 | 0.216 | 85.221 | 5.305 |
| Ground Truth | — | — | — | 5.398 |
| Ours | 0.557* | 0.209* | **87.677** | 5.407 |

The **bold** text denotes the best result and the second-best results are denoted with *.

**Table 2: Time and memory consumption of image synthesis**

| | SD | SD w. LoRA | Adapter | ControlNet | Ours |
|---|---|---|---|---|---|
| Time | 8.13it/s | 7.70it/s | 8.31it/s | 5.51it/s | 7.93it/s |
| Parameters | 4067MB | 4080MB | 4362MB | 5445MB | 4668MB |

truth and synthesized images, where higher scores denote better alignment. Similar to the CLIP image, the CLIP-aesthetic predictor applies CLIP embeddings with an MLP layer to predict the average preference for an image. Higher scores indicate better results.

## 4.2 Comparison Results

**Quantitative Comparison:** Table 1 illustrates the quality of synthesized images between our method and other state-of-the-art methods. For diffusion-based models, our proposed method outperforms the others in terms of the LPIPS scores. The FID score of the images from our method also achieves the best results in diffusion models, which means the generated images are of higher quality than those from other methods. CycleGAN achieves favorable results on these two criteria by introducing only minor changes, though it does not fully capture the style of the illustrative image. This will be discussed in more detail in the User Study section. CLIP-image is a criterion that evaluates the quality of the synthesized images; our method performs better than the others, indicating that it carries more of the illustrative style. For CLIP-aesthetic, the score from our method is higher than that of Adapter and CycleGAN but lower than those of ControlNet, LoRA, and SD with text. The reason for this is that this criterion is derived from feature maps based on the pretrained CLIP model on the LAION-5B dataset, which contains a larger proportion of real images. The scores are assigned based on these real images, which can lead to a shift in scoring. On the other hand, the scores obtained by our method are closer to that of the ground truth, indicating that the synthesized images from our method more closely resemble the ground truth compared to those from other methods.

Table 2 denotes the details of consumption. Image synthesis tests were conducted on a single RTX 3090 GPU with the resolution of the synthesized images set to 512×512 pixels. The model parameter sizes are calculated based on a float32 precision format. As shown in the table, the inference time of our method is significantly shorter than that of ControlNet's, and slightly longer than those of the Adapter and the baseline model. However, the usage of the time is comparable. In terms of memory usage for parameters, our method requires slightly more memory than the basic Stable Diffusion and T2I-Adapter, but much less than ControlNet. This is because the style adaptation module in our method has far fewer parameters

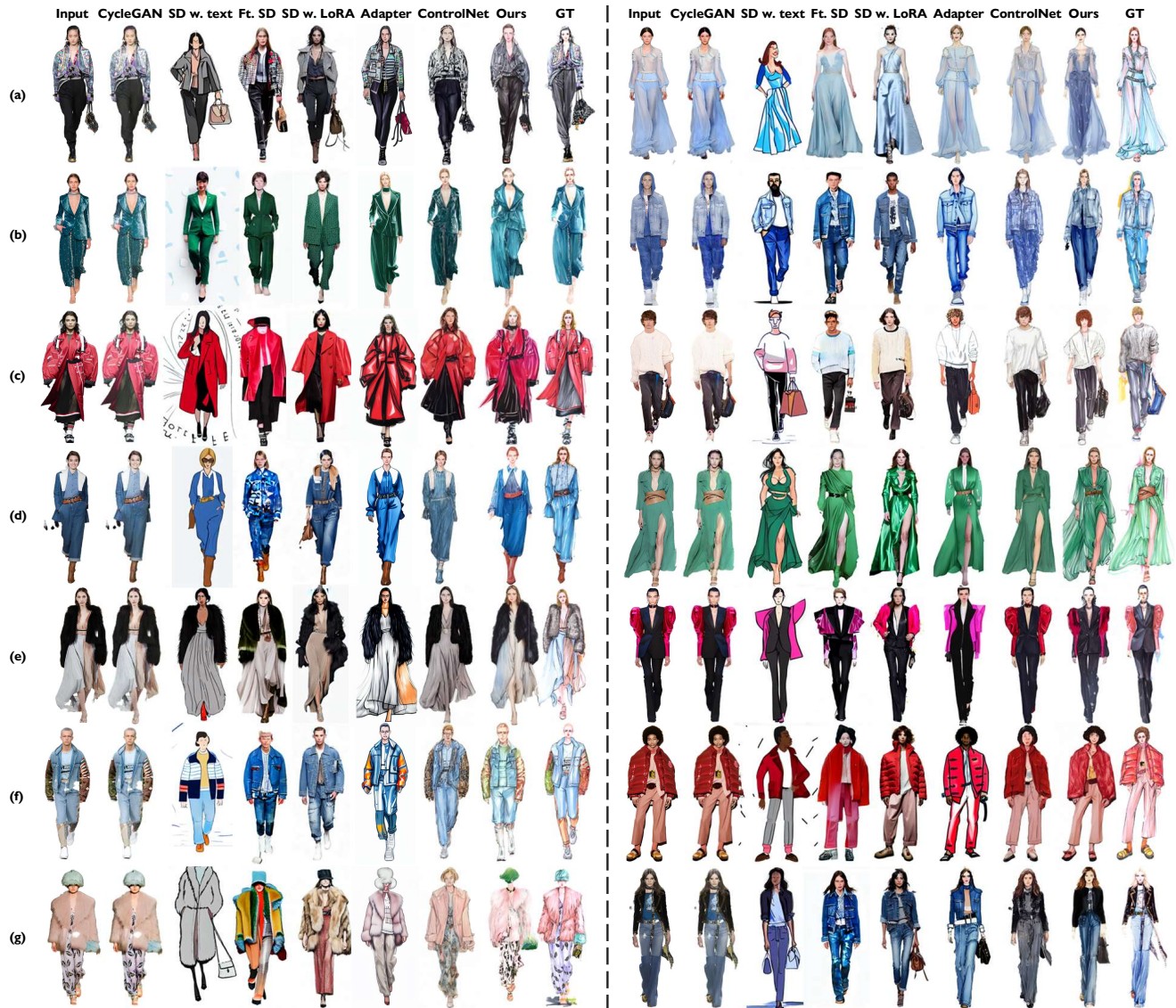

**Figure 5: Qualitative comparison between Uni-DlLoRA and other state-of-the-art approaches. From left to right, the displayed results correspond to CycleGAN, Stable Diffusion (SD), fine-tuned SD, SD with LoRA, T2I-Adapter, ControlNet, and our method, respectively. The text caption is utilized for content synthesis in SD, fine-tuned SD, and SD with LoRA, while the prompt 'illustrative style' is used for style guidance in both SD and fine-tuned SD. The figure is best viewed when zoomed in.**

than the extra UNet Denoiser. In summary, our proposed method requires only a small amount of extra memory compared to Stable Diffusion and can generate high-quality images with an illustrative style on a home-use GPU.

**Qualitative Comparison:** The generated results include Cycle-GAN [54] for GAN-based models, and for diffusion-based models, we have pretrained Stable Diffusion (SD), Fine-tuned SD [36], SD with LoRA [13], ControlNet [50], and T2I-Adapter [30], along with results from our method for comparison. Pretrained Stable Diffusion (SD) has zero-shot capabilities but cannot perform style transfer independently; text prompts are adopted for its synthesis. Similarly, prompts are also adopted for Fine-tuned SD and SD with LoRA.

Figure 5 illustrates the comprehensive qualitative comparison. Generally speaking, images generated by T2I-Adapter, ControlNet, and our method are able to capture the illustrative style, while Cycle-GAN and SD with LoRA struggle to alter the style of the source image. Since the pretrained SD learns the illustrative style from a universal dataset, it cannot accurately capture the specific illustrative style of a real designer. Specifically, all methods can preserve the appearance of the input image in each row. However, results from CycleGAN struggle to modify the style of the images, whereas the generated images capture the style of the real images and appear more realistic when compared with illustrative images. The images generated from fine-tuned Stable Diffusion, Stable Diffusion with text, and Stable Diffusion with LoRA are able to capture

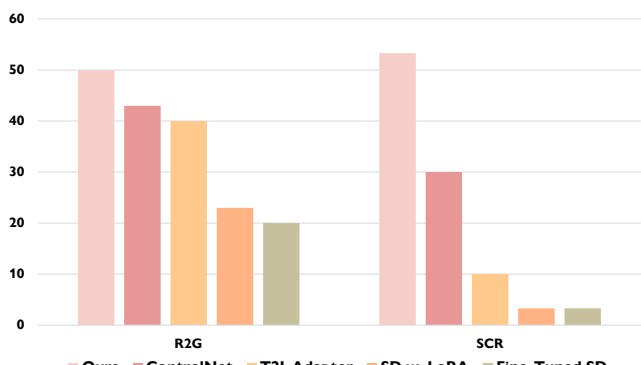

Figure 6: User study. The table on the left represents the R2G score, while the table on the right illustrates the SCR score.

the semantic information from the input images. However, they lack the detailed nuances of the illustrative style. Additionally, the generated images do not integrate harmoniously with the overall composition. For instance, see rows (b) and (c): the resulting images appear rigid and exhibit a discernible style conflict when compared with the ground truth. In terms of the generated results from the T2I-Adapter, ControlNet, and our own model, all are capable of conveying the illustrative style while maintaining the appearance of the runway model. However, the T2I-Adapter and ControlNet may fall short in replicating the intricacies of the clothing. For example, there is a slight color shift in the results from the T2I-Adapter evident in rows (b) and (d). Additionally, the clothing details exhibit variations in row (c). As for the images generated by ControlNet, while they effectively capture the style and general appearance, there is potential for improvement in clothing details, such as the red attire in row (c) and the gray clothing in row (a).

**User Study:** Since the evaluation of illustrations is often abstract and subject to many human perceptions, the opinions of 100 human participants will be used as the standard for assessing effectiveness. A user study was conducted to assess the abstract quality of the results from our method compared to those obtained by other methods. Two approaches are adopted for this evaluation. The first employs R2G metrics, as mentioned in the research by Zhu et al. (2019) [57], which measures the percentage of generated images classified as ground truth (illustrative images). The second criterion involves the scores assigned to the highest-quality results by the participants. They are instructed to base their evaluations on the ability of each competing approach to produce accurate clothing and an illustrative style. This is quantified using another metric named *SCR*, defined as the percentage of images considered the best among all the models. Higher values in these three metrics indicate better performance. The comparative results of the study are illustrated in Figure 6, which clearly demonstrates that our methods surpass the others in terms of human perception: 50% of the results from our method are perceived as ground truth. Regarding the SCR metric, our *SCR* score is 53%, indicating that participants favored our approach more frequently than the competing methods.

### 4.3 Ablation Study

An ablation study was conducted to evaluate the impact of each component within the proposed model in the StylishU-SR dataset. Table 3 illustrates the impact of each component on the dataset.

**Table 3: Qunatitative comparison between each component.**

| Methods | Metrics | | | |
|---|---|---|---|---|
| | $FID_{64}$ ↓ | $LPIPS$ ↓ | CLIP-image ↑ | CLIP-aes ↑ |
| Baseline (SD) | 2.677 | 0.298 | 74.748 | **5.598** |
| Uniadapter | 0.814 | 0.214 | 83.789 | 5.291 |
| Uni-SgLoRA | 0.749* | 0.213* | 84.814* | 5.356 |
| Full Model | **0.557** | **0.209** | **87.677** | 5.407* |

The **bold** text denotes the best result and the second-best results are denoted with *

| Real Image | Ground Truth | Baseline | Uniadapter | Uni-SgLoRA | Ours | Reconstructed |

**Figure 7: Ablation results on the StylishU-SR. The images in this figure correspond to the ablation studies in Table 3.**

Baseline (SD) neither employs the uniadapter module only a UNet-based noise prediction module with extra prompt "illustrative style". Although it can generate images with a precise appearance in Figure 7, its ability to retain the illustrative style and preserve the texture of the garments is limited. To effectively model the complex textures within the clothing, a learnable adaption module extracts image information and then sent to the UNet denoiser. When incorporating latent appearance features extracted by a pretrained adaptation module, we refer to this process as Uniadapter. Compared to the baseline, the Uniadapter reduces the $FID_{64}$ score from 2.677 to 0.814, indicating a performance improvement. As shown in Figure 7, the results from Uniadapter capture more appearance and image information than the baseline model. To enhance the style translation, a style adaptation module is adopted during both training and sampling to capture the style features. From the table, it is clear to see that the style adaption module improves the results in all four criteria. The SgLoRA module not only improves the generation quality from the statistics but also in human perception shown in Figure 7. To further disentangle the style and content information of the source image, the Dual-LoRA module is adapted to align the output image content and style with the source image content and style. The last column in Figure 7 illustrates that our model can successfully catch the content and reconstruct the source image with good quality. In comparison with Uni-SgLoRA, our full model improves the $FID_{64}$, $LPIPS$, $CLIP-image$ and $CLIP-aes$ by a margin of 0.192, 0.004, 2.863 and 0.051, respectively.

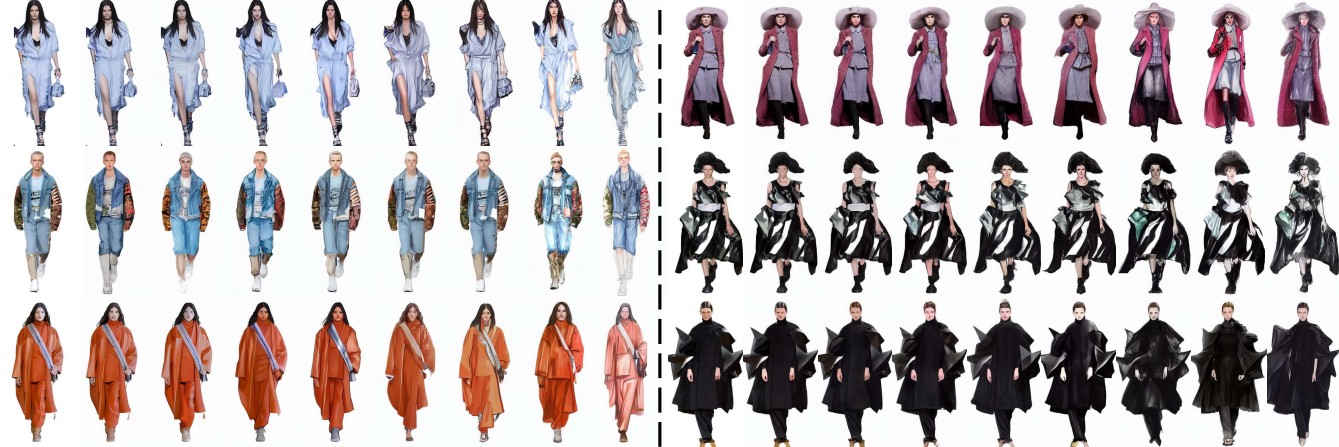

Original — — — — — — — — —⟶ Illustrative    Original — — — — — — — — —⟶ Illustrative

**Figure 8: Style interpolation. Images are synthesized by combining a source image with varying style strengths ranging from 0 to 1. Generated images progressively carry the illustrative style. Both images inside the dataset and in the wild are evaluated.**

## 4.4 Illustrative Style Interpolation

The proposed model is capable of modifying the final generated graphic's illustrative style by adjusting the sampling positions. We utilize the DDIM [43] sampling approach for the generation task. Specifically, the image generation task involves sampling a total of 50 times. To adjust the strength of the effect, we perform linear interpolation with values between 0 and 1. Based on this value, we add Gaussian noise of corresponding strength to the latent features extracted from the input image. Additionally, the introduction of Gaussian noise at various timesteps is also based on the interpolated value. This allows the generated image to obtain more style information. Three samples from the test dataset and three real-world samples are selected to demonstrate the effectiveness of the illustrative style interpolation. As depicted in Figure 8, it is evident that the style of the images undergoes a gradual transformation from the left source image to the right source image. This gradual shift showcases the model's capability to provide a smooth transition in two different styles.

## 4.5 Generate Image in the Wild

Our model, which is fine-tuned based on a pretrained stable diffusion model, exhibits strong robustness and is also capable of performing illustrative style transfer on runway images outside of the dataset. The images in the figure showcase some successful instances of illustrative style transformation. As illustrated in the right half of the Figure 8. we can observe that for the image that is out-of-dataset, the method not only generates images that carry illustrative style but also captures varying degrees of style based on the number of sample steps. This successful style evolution and the consistency observed in images from the test dataset and images in the wild underscore the robustness and strong adaptability of the proposed method.

## 5 LIMITATIONS

Although our proposed method achieves solid results in most cases, it still fails in certain scenarios as shown in Figure 9. For instance, due to the images being formed by the overlay of noise, precise

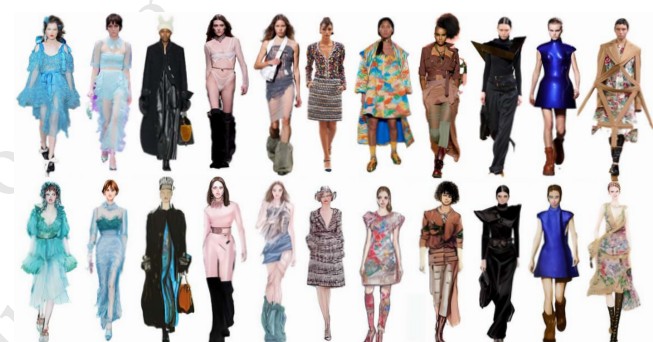

**Figure 9: Failure cases using the proposed method.**

alignment remains an area in the complicated domain like fashion for improvement. As demonstrated in the figure, the generated illustrative images still exhibit noticeable differences from the original in aspects such as the texture of the clothing (the first six examples), and the shape of the garments (the rest six examples). The aforementioned examples also prove that the image transformation through this method entails a certain level of randomness and does not align as closely with the source image as might be desired. The sketch images may be insufficient to carry all the detail necessary, thus failing to constrain the final image synthesis adequately.

## 6 CONCLUSIONS AND FUTURE WORK

Leveraging the existing challenges within illustrative transformation, this paper has created a new high-resolution real-to-illustration dataset. It also introduces a novel approach to resolve these challenges. The proposed model incorporates the concept of disentanglement, utilizing a shared image extractor and distinct style adaption modules to learn the content and style of images, and converts these into an illustrative style. This innovation contributes significantly to the fashion field. Nevertheless, the method has limitations, and the illustrative style transformation does not fully achieve alignment with the source images. In the future, we aim to achieve complete content alignment while better-capturing texture information and further enhancing the style transformation.

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

Received 20 February 2007; revised 12 March 2009; accepted 5 June 2009

