# OpenReview forum: "Uni-DlLoRA: Style Fine-Tuning for Fashion Image Translation"
_acmmm.org/ACMMM/2024/Conference — MM2024 Poster_

### Official Review · Reviewer_dSn6 · 2024-05-15

**Rating:** 4
**Confidence:** 3

**Summary:**

This paper introduces a novel model called Uni-DlLoRA, which incorporates the original images within a pre-trained diffusion-based model, leveraging the innovative Uni-adapter extractors. It also utilizes the DualLoRA module to provide distinct style guidance, thus enhancing generative capabilities while minimizing the need for additional parameters. Furthermore, a new multimodal dataset with higher-quality images and accompanying captions is introduced, which is built upon an existing real-to-illustration dataset.

**Strengths:**

(1) The performance of this paper is good. Satisfactory visual results can be produced using the proposed method.

(2) The proposed method is novel and interesting. It applies a Uni-adapter to extract latent features from input images and enhances learning in fashion image translation through the novel Dual-LoRA module.

(3) A new dataset is constructed, which is beneficial for this research field.

(4) Extensive ablation studies have been conducted to verify the effectiveness of the design components.

**Limitations:**

(1) Missing comma after all formulas.

(2) The text below Table 3 is blocked by Figure 7.

(3) FLOPs or GFLOPS is expected to evaluate the computational cost.

(4) In the Introduction section, the motivation for the proposed Uni-DlLoRA method is not explicitly stated, which makes the paper not so academic. The authors simply illustrate their proposed method, but do not explain the inspiration and principle of the proposed method.

(5) The visual results are shown in small sizes. We suggest zooming in on a squared region of the image.

(6) Why not compare with StylishGAN mentioned in the introduction section?

**Suitability:**

3

---

### Official Review · Reviewer_aAqh · 2024-05-26

**Rating:** 4
**Confidence:** 3

**Summary:**

The paper presents a novel approach to image-to-image (i2i) translation specifically tailored for the fashion industry. It introduces a model called Uni-DlLoRA that focuses on real-to-illustrative style transfer for fashion images. The model integrates original images within a pretrained diffusion-based framework, enhanced by novel Uni-adapter extractors and a Dual-LoRA module to provide distinct style guidance. This method not only optimizes generative capabilities but also minimizes the addition of extra parameters. Additionally, the paper introduces a new multimodal dataset that features high-quality images with captions, aimed at addressing existing limitations in fashion image datasets.

**Strengths:**

The Uni-DlLoRA model introduces innovative components like Uni-adapters and Dual-LoRA modules, which are novel in their application to fashion i2i translation. The integration of these elements into diffusion models showcases a robust approach to preserving identity and style distinctiveness in generated images.


The paper provides extensive qualitative and quantitative evaluations, demonstrating the superiority of the proposed method over existing techniques. The introduction of a new high-quality dataset also strengthens the evaluation, providing a benchmark for future research.

The paper introduces a high-quality multimodal dataset designed specifically to overcome existing limitations in the fashion domain. This dataset features images with enhanced resolution and precise textual descriptions, offering a robust resource for advancing research and application in fashion image synthesis and translation.

**Limitations:**

The paper lacks detailed explanations of the experimental setup, including hyperparameters, training duration, computational resources used, parameter settings, layer configurations, and optimization techniques, making it challenging to replicate the experiments precisely.

Conduct additional experiments comparing the proposed method with the latest state-of-the-art baselines to provide a more comprehensive evaluation of its performance.

Validate the approach in different contexts beyond fashion to showcase its generalizability and potential applications in various fields.
While the new dataset is a strength, the model's performance heavily relies on the quality and characteristics of this specific dataset. The generalizability of the model to other datasets or broader real-world applications might be limited.

**Suitability:**

3

---

### Official Review · Reviewer_FJjq · 2024-05-28

**Rating:** 4
**Confidence:** 2

**Summary:**

In this paper, the authors explore the challenge of translating fashion images between the realms of realism and illustration. The authors introduce a model, Uni-DlLoRA, which utilises a pretrained diffusion-based generative model augmented by custom modules for style guidance and feature extraction. A significant enhancement is the development of a high-quality multimodal dataset with improved resolution images and curated captions. The model aims to optimize image generation while ensuring stylistic accuracy and diversity.

**Strengths:**

- The creation of a new dataset with high-resolution images and accurate text annotations addresses the common problem of dataset quality in fashion image synthesis.
- It seems that the paper provides comprehensive validation of the model through both quantitative metrics and qualitative analysis.
- Code is provided in the supplement.

**Limitations:**

- While the contributions on the refined dataset are good, the technique contribution of this paper seems not that strong, which is essentially an application of diffusion model on fashion image translation. The style and content disentanglement have also been introduced in existing diffusion-based works. The authors are encouraged to further justify their technical contributions.

- There are currently too few details for the user study. How did the authors arrange the interface shown to the participants? Were the images from different methods shuffled randomly for each show?

- Minor: In Tab. 3, the footnote below the table overlaps with Fig. 7.

**Suitability:**

3

---

### Meta-Review · Area_Chair_pZHh · 2024-07-01

**Recommendation:** Accept (Poster)
**Confidence:** 5

**Metareview:**

This paper has two contribution. First, they proposed a novel image-to-illustration translation model called Uni-DILoRA. Second, they constructed a new dataset for this purpose.
Experimental results show that the proposed method is superior to the baseline methods.
All the reviewers are satisfied with the authors' responses.
Therefore, I would recommend acceptance of this paper.

Just a few comments to improve this paper:
In Table 1, "Ours" can be "ours".
The caption for Table 3 is overlaped with Fig. 7